# The Potential of FGF-2 in Craniofacial Bone Tissue Engineering: A Review

**DOI:** 10.3390/cells10040932

**Published:** 2021-04-17

**Authors:** Anita Novais, Eirini Chatzopoulou, Catherine Chaussain, Caroline Gorin

**Affiliations:** 1Pathologies, Imagerie et Biothérapies Orofaciales, Université de Paris, URP2496, 1 rue Maurice Arnoux, 92120 Montrouge, France; anitarsnovais@gmail.com (A.N.); eirini.chatzopoulou@gmail.com (E.C.); catherine.chaussain@u-paris.fr (C.C.); 2AP-HP Département d’Odontologie, Services d’odontologie, GH Pitié Salpêtrière, Henri Mondor, Paris Nord, Hôpital Rothschild, Paris, France; 3Département de Parodontologie, Université de Paris, UFR Odontologie-Garancière, 75006 Paris, France

**Keywords:** FGF-2, bone tissue engineering, angiogenesis, mineralization, signaling pathways

## Abstract

Bone is a hard-vascularized tissue, which renews itself continuously to adapt to the mechanical and metabolic demands of the body. The craniofacial area is prone to trauma and pathologies that often result in large bone damage, these leading to both aesthetic and functional complications for patients. The “gold standard” for treating these large defects is autologous bone grafting, which has some drawbacks including the requirement for a second surgical site with quantity of bone limitations, pain and other surgical complications. Indeed, tissue engineering combining a biomaterial with the appropriate cells and molecules of interest would allow a new therapeutic approach to treat large bone defects while avoiding complications associated with a second surgical site. This review first outlines the current knowledge of bone remodeling and the different signaling pathways involved seeking to improve our understanding of the roles of each to be able to stimulate or inhibit them. Secondly, it highlights the interesting characteristics of one growth factor in particular, FGF-2, and its role in bone homeostasis, before then analyzing its potential usefulness in craniofacial bone tissue engineering because of its proliferative, pro-angiogenic and pro-osteogenic effects depending on its spatial-temporal use, dose and mode of administration.

## 1. Introduction

The skeletal system is dynamic, metabolically active and functionally diverse. As well as a structural function, it has a metabolic role. It supports and protects the vital internal organs and is the site of synthesis of the hematopoietic marrow and provides sites of muscle attachment for locomotion. Bone is involved in both mineral metabolism, via calcium and phosphate homeostasis, and acid–base balance, via the buffering of hydrogen ions [1]. Moreover, it has been suggested that bone has other important endocrine functions in fertility, glucose metabolism, appetite regulation and muscle function [2,3]. The craniofacial area is prone to trauma and pathologies that often result in large bone damage, these leading to both aesthetic and functional complications for patients. Throughout life, the craniofacial area is at risk of complex injuries that require bone grafting to restore function. The etiology of such injuries may be accidental (e.g., acute trauma), congenital (e.g., birth defects or deformities), pathological (e.g., maxillofacial tumors, such as ameloblastoma, or infection) or surgical. Whenever the lesions are extensive, they cause large bone defects that cannot self-repair because they exceed the body’s natural regenerative capacity. The “gold standard” for treating these large defects is autologous bone grafting. Since the defects are extensive, harvesting of the necessary bone at the donor site can cause major morbidity, including bone deformity, as well as pain and occasionally continuous progressive resorption. Tissue engineering has made it possible to approach these issues from another angle. Before talking about tissue engineering; however, it is necessary to describe the tissue of interest. This review is, therefore, organized into three sections, the first describing general bone physiology (e.g., its composition, functioning and signaling pathways), the second the implication of FGF-2 in bone physiology, and the third highlighting the interesting characteristics of FGF-2 for craniofacial bone engineering and its various current potential uses for promoting bone repair.

## 2. Background on Bone Physiology

In the skeleton, there are two types of bone differentiated by their structure—cortical and trabecular. Although both types have an identical chemical composition, they differ both macroscopically and microscopically [4] (Appendix A
Figure A1).

### 2.1. Bone Composition

Bone is a mineralized connective tissue composed of bone cells, vessels, and an extracellular matrix (ECM), which is produced by the bone cells. The proportion of these components varies with bone type (long or flat bone) and anatomical site [5]. The mineralized component of bone tissue gives rigidity and hardness to the material, while the organic components of the ECM provide flexibility [6].

#### 2.1.1. Bone Cells

The osteoblast, a specialized bone-forming cell, mostly differentiated from mesenchymal stem cells (MSCs) [7,8], produces and secretes the major bone matrix protein essential for matrix mineralization, namely, type I collagen [9,10]. These cells work in clusters on the bone surface [5], becoming committed to one of various possible fates, thanks to some key proteins and pathway signaling, such as runt-related transcription factor 2 (Runx-2) and Osterix (Figure 1).

Osteocytes, the most abundant cell type, are found in mature bone and are long-lived [11]. They are distributed throughout the bone matrix and can interact with other osteocytes or osteoblasts on the bone surface, vasculature and bone cells on the surface of bones, in a complex intercellular network [6,12,13]. Osteocyte-transduced signals seem to orchestrate bone response by regulating the synchronized action of osteoblasts and osteoclasts [14,15,16] (Figure 1).

The osteoclasts, multinucleated cells formed by the fusion of precursors derived from the hematopoietic cells of the mononuclear lineage [17,18], are the only cells known to be capable of resorbing bone (Figure 1).

Other types of cells, such as chondrocytes are found within the bone. These are derived from pluripotent MSCs, and secrete type II collagen, participating in the endochondral ossification process [19].

#### 2.1.2. Bone Extracellular Matrix

Bone ECM contains both a mineral and an organic phase, the latter representing approximately 90% of the organic content of bone tissue. The mineral portion is largely calcium phosphate in the form of hydroxyapatite crystals deposited in an osteoid matrix and is responsible for bone’s mechanical rigidity and load-bearing strength. The organic portion, which contains water, non-collagenous proteins, lipids and specialized bone cells, gives flexibility and elasticity to bone tissue [1,6].

Non-collagenous proteins constitute 10% to 15% of total bone protein content. Almost 25% of non-collagenous proteins are adsorbed from serum and due to their acidic properties are able to bind to the matrix [20], which allows strengthening of the collagen structure and regulating its mineralization [4]. Osteocalcin is the major non-collagenous protein, and is involved in calcium binding, stabilization of hydroxyapatite in the matrix and the regulation of bone formation, acting as a negative regulator, inhibiting premature or inappropriate mineralization [21,22].

### 2.2. Bone Formation

Bones form through two different processes: bone modeling and bone remodeling (Figure 2). Bone modeling occurs primarily during growth and development in childhood, although it can also appear after the skeleton has matured [23,24].

In contrast, bone remodeling occurs after the skeleton has reached maturity, during adulthood, involving resorption of old or damaged bone, and its replacement by newly formed bone. Here, we focus on bone remodeling.

#### 2.2.1. Bone Remodeling Process

Bone remodeling is essential for structural integrity, biomechanical stability, bone volume and calcium/phosphate homeostasis [1,4,25]. In normal adult bone, there is a homeostatic balance in which bone resorption is followed by bone formation for maintaining bone strength and mineral homeostasis, keeping the overall bone volume and structure unchanged [26,27]. By this process, about 10% of the skeleton may be renewed every year [28]. That is, bone remodeling only takes place when it is required, either because the specific area is damaged and/or old.

The bone remodeling cycle has several phases: cell activation, bone resorption, reversal, bone formation and termination (Figure 2).

#### 2.2.2. Regulation of Bone Remodeling

The remodeling cycle is tightly regulated to achieve a balance between bone formation and resorption. Considering that remodeling can occur in several locations simultaneously, local regulation is critical to achieving this balance.

Major signaling pathways

1RANKL/RANK/OPG

One of the major signaling pathways that regulates bone remodeling involves three proteins: RANKL, receptor activator of nuclear factor kappa-B (RANK) and osteoprotegerin (OPG). The interaction between these proteins determines whether, at a specific location, bone resorption or bone formation occur. RANKL is a cytokine expressed on the surface of osteoblasts, osteocytes and chondrocytes. It activates nuclear factor kappa-B (NF-kB) and other signaling pathways through the interaction with its receptor, RANK, located on osteoclast precursors. RANKL/RANK activation has an important role in osteoclast differentiation, allowing osteoclast formation, activation and survival [13,29,30,31,32]. OPG, a soluble decoy receptor for RANKL expressed by osteoblasts and osteocytes, binds to RANKL with high affinity, preventing it from binding to its receptor RANK. Thus, OPG is a natural inhibitor of RANKL. The RANKL/OPG expression rate regulates the extent of osteoclast formation and activity [33,34,35] (Figure 1).

2.Wnt signaling

Wingless-related integration site (Wnt) molecules are cysteine-rich glycoproteins involved in controlling cell proliferation, cell-fate specification, gene expression and cell survival. Wnt signaling pathways are involved in bone formation, having an anabolic effect, and increasing bone density and strength, by regulation of osteoblast differentiation and function [36,37] (Figure 1).

Wnt pathways also play a major role in osteoclast differentiation. Specifically, Wnt canonical signaling up-regulates OPG and downregulates RANKL, which inhibits osteoclast formation and therefore bone resorption [36,38]. In contrast, activation of the noncanonical pathway in osteoclast precursors enhances RANKL-induced osteoclastic differentiation [36].

Wnt signaling is inhibited by secreted proteins such as sclerostin and Dickkopf-related protein 1–4 (DKK-3/4) synthesized by osteocytes [39,40] (Figure 1). Nevertheless, during bone remodeling, osteocytes decrease the expression of sclerostin and Dickkopf-related protein 1 and 2 (DKK-1/2), allowing osteoblast bone formation to occur after bone resorption. After the completion of remodeling, newly formed osteocytes become entombed within the bone matrix and start expressing Wnt inhibitors, stopping further bone formation [12].

Endocrine regulation

Bone turnover by osteoblasts and osteoclasts is essential for the maintenance of bone strength and morphology. Due to its importance, this process must be thoroughly regulated to prevent malfunction of the remodeling process at any stage of the cycle. Several hormonal agents are involved in this regulation, including PTH [10], 1,25 (OH) Vitamin D [4], calcitonin [41], thyroid hormone [42], growth hormone [43], glucocorticoids [44] and sex hormones [45].

Paracrine regulation

Some cytokines may have stimulatory and inhibitory effects on bone metabolism. Cytokines like interleukin 1 (IL-1), IL-6, and tumor necrosis factor (TNFα) can increase osteoclastic resorption, whereas others, such as interleukin 4 (IL-4) and gamma interferon, decrease osteoclast proliferation and differentiation [46,47].

Prostaglandins may also influence bone formation, although their exact role remains unclear. Prostaglandin E_2_ (PGE_2_) is a major inducer of bone resorption and is thought to increase the RANKL/OPG ratio to improve osteoclastogenesis (Figure 1). At the same time, it has been hypothesized to stimulate osteoblast proliferation and differentiation thereby enhancing bone formation [48]. Inside the bone matrix, there are growth factors that affect bone metabolism. The main families involved are the transforming growth factor β (TGFβ) family (TGFβ and BMPs) and the fibroblast growth factor (FGF) family.

### 2.3. Bone Vascularization

There is an intimate link between osteogenesis and angiogenesis, both processes needing to be tightly coupled for optimal physiological bone function [49]. In the event of a critical bone defect, early vascularization is necessary for osteogenic reconstruction, to allow the nutritional support for the bone grafts [50,51,52]. The close relationship between blood vessels and bone cells is also well illustrated by abnormalities resulting from inappropriate vascularization, these leading to the appearance of skeletal malformation, such as craniofacial dysmorphology [53].

In the close connection between these two processes, several factors have been described as being both angiogenic and osteogenic. Notably, it has been demonstrated that hypoxia (oxygen tension) and the vascular endothelial growth factor (VEGF) family affect endochondral angiogenesis as well as cells from the bone lineage [54,55,56].

Indeed, several pro-angiogenic factors are involved in bone repair. Some of these factors have direct effects, both having angiogenic properties and regulating osteogenic molecules, like BMPs, angiopoietin (Ang), platelet-derived growth factor (PDGF) and insulin-like growth factor (IGF) family members (Appendix A
Table A2).

Others are well known to indirectly enhance bone repair using their pro-angiogenic properties. VEGF, an endothelial cell (EC)-specific mitogen, is secreted by cells involved in skeletal development and repair, such as hypertrophic chondrocytes or differentiating mesenchymal cells, osteoblasts, and ECs [57,58]. It can be a chemoattractant molecule, engaging ECs into bone tissue and tightly controlling the differentiation and functions of osteoblasts and osteoclasts [55,59,60,61,62]. It is also involved both in endochondral ossification promoting vessel invasion and cell recruitment [59,63], and in intramembranous ossification, by affecting bone cell activity [52,53,64]. It has been reported that VEGF upregulates the RANK receptor in ECs and strongly stimulates angiogenesis [65]. In turn, RANKL may have an important role in enabling EC survival via the phosphatidylinositol 3-kinase/protein kinase B (PI3K/Akt) signaling pathway [66,67], one of the major signaling pathways triggered by activated VEGF receptor [68] (Figure 1).

The hypoxic signaling pathway has been reported to directly enhance VEGF expression, being described as a major regulator of VEGF expression [69,70]. Indeed, hypoxia-inducible transcription factors (HIFs) are expressed in osteogenic cells, notably osteoblasts, and hence, hypoxia upregulates VEGF expression, thereby promoting angiogenesis and osteogenesis. Thus, hypoxia and VEGF signaling are involved in the coupling of angiogenesis and osteogenesis [54,55,71].

This review focuses on the FGF family and specifically fibroblast growth factor 2 (FGF-2) and its potential usefulness in bone tissue engineering.

## 3. FGF-2 in Bone Homeostasis

### 3.1. FGF/FGFR Signaling in Bone

Various growth factors act within the bone matrix and influence bone metabolism. The main families involved are the TGFβ (TGFβ and BMPs) and FGF families. The latter is also involved in angiogenesis, which makes it interesting to study more in detail. In this review, we focus on the great potential of FGF-2 in bone metabolism and discuss the interest in its use in bone tissue engineering.

Members of the FGF family are single-chain polypeptide growth factors of approximately 20–35 kDa. At least 23 members of this family have been described in mammals [72], from FGF-1 to FGF-23. The secreted FGFs are differentially expressed in almost all tissues of the developing embryo, functioning as essential regulators of the earliest stages of embryonic development. They are also expressed in postnatal and adult tissues, fulfilling essential roles in tissue homeostasis, repair, regeneration, angiogenesis and bone metabolism [61,73,74,75,76,77,78]. There are four FGF receptors: FGFR-1 to FGFR-4 [79,80].

Given the ubiquitous roles of FGF signals in development, homeostasis, and disease, tight regulation of the pathways is essential through cofactors that participate in the affinity and specificity of FGFR-binding, and also in specifying signaling activities [81,82]. The endocrine regulation of FGFs, including FGF-19, FGF-21 and FGF-23, is mediated by their binding with the Klotho protein family (α or βKlotho or KLPH) [83,84,85], which allows them to circulate through the matrix without being trapped and stored [82], thereby acting in an endocrine-like fashion.

Canonical FGFs exert their pleiotropic effects by binding and activating the FGFR subfamily of receptor tyrosine kinases that are coded by four genes (FGFR-1, FGFR-2, FGFR-3, and FGFR-4) in mammals. [86]. The four FGFRs have distinct ligand specificity and are expressed in a tissue-specific manner [87]. Cofactors such as heparan sulfate and klotho are low-affinity receptors that do not induce a biological signal but rather are used as auxiliary proteins to regulate FGF binding and subsequent phosphorylation of adaptor proteins for four major intracellular pathways [88,89,90,91]: rat sarcoma-mitogen-activated protein kinase-extracellular signal-regulated kinase 1 and 2 (RAS-MAPK-ERK1/2), PI3K-AKT-glycogen synthase kinase 3 (GSK3), phospholipase Cγ-protein kinase C (PLCγ-PKC), and signal transducer and activator of transcription proteins-Janus kinase (STAT-Jak) [81,86,92,93]. The main regulation pathway remains the paracrine one, involving FGF-2, which binds FGFRs through a heparan sulphate glycosaminoglycan binding site, limiting their diffusion through the ECM [88,90,94]. Upon binding of FGF to its receptor, receptor dimerization and transautophosphorylation of the kinase domain take place [95] (Figure 3).

FGFs can activate several MAPKs such as C-Jun N-terminal kinase (JNK), ERK, and p38MAPK [96], that share many structural similarities, while inducing different responses [97,98]. The ERK1/2 branch, one of the major routes for FGF signaling [99], promotes a mitogenic response and is observed in all cell types, while p38 and JNK kinase are usually associated with inflammatory and stress responses [98,100] and are mainly involved in the cell cycle, cytoskeleton and cell migration [101]. For instance, FGF-2 regulates mesenchymal stem cell migration [102] via the PI3K-AKT pathway, neural progenitor cell proliferation via PI3K/GSK3 signaling [103], and fibroblast migration via PI3 kinase, Ras-related C3 botulinum toxin substrate 1 (Rac1) and JNK [104], as well as promoting bone marrow MSC proliferation through the ERK1/2 pathway [105]. Activation of the MAPK pathway, in response to FGF-2 signaling, is key in determining the activity of RUNX-2, a master transcription factor of bone formation [106]. FGF-2 promotes osteoblast proliferation and differentiation [107], through the activation of the ERK1/2 pathway [108,109,110] and a crucial role of this pathway has been identified in the differentiation of osteoblasts and chondrocytes [107,111,112,113]. Specifically, MAPK phosphorylates Runx-2 Ser/Thr residues, a critical step for Runx2 acetylation and stabilization against degradation [114]. Sprouty (SPRY) and Sprouty related (SPRED) are two antagonists of this pathway [115,116] that act as intracellular antagonists of FGFR signaling, the first by suppressing ERK1/2 activation [117] and the second by suppressing rapidly accelerated fibrosarcoma (Raf) activation [118] (Figure 3).

The PI3K/AKT pathway is implicated in cell polarization, migration, cell fate determination and apoptosis. For instance, FGF-2 enhances the migratory activity of periodontal ligament cells (PDLSCs) through this pathway. In addition, FGF-2 induced Akt phosphorylation promotes proliferation in neural progenitor cells [103] and MSCs [105]. Sprouty2 has also been associated with the PI3K/Akt pathway, suppressing Akt phosphorylation [101].

The FGFR tyrosine kinase domain can also directly phosphorylate PLCγ, which leads to the hydrolysis of phosphatidylinositol 4,5-bisphosphate to produce inositol triphosphate (IP3) and diacylglycerol. Subsequently, IP3 increases intracellular calcium ion levels, while diacylglycerol activates PKC. Furthermore, it has been shown that FGFs can induce expression of receptor of activated protein C kinase 1 (RACK-1), a protein that further stabilizes activated PKC [119]. It has been shown that FGFR-1 and FGFR-2 can directly bind to activate PLCγ1 [120]. In fact, FGF-2-induced activation of the PLCγ pathway is involved both in increased expression and transcriptional activity of RUNX2 [121]. In dental tissue-derived mesenchymal stem cells, it has been shown that neurogenic differentiation in human dental pulp stem cells (DPSCs) is induced by FGF-2 via the PLCγ signaling pathway [122], which is already known for its role in the differentiation of neuronal cells [123].

Other pathways such as STAT [96] and ribosomal S6 kinase 2 (RSK2) pathways [124] have been shown to be involved in FGF-2 signaling. The activated FGFR also phosphorylates and activates STAT1, STAT3, and STAT5, to regulate STAT pathway target gene expression [79]. These pathways control the steps of osteoblastogenesis resulting in the modulation of bone cell proliferation, differentiation, and apoptosis, depending on the stage of cell differentiation [125,126]. Therefore, normal FGFR activity is critical for the development of numerous types of tissue, including the craniofacial skeleton, as observed in several genetic diseases causing abnormalities in bone and cartilage formation due to mutations in the genes encoding FGFs or their receptors [127,128].

### 3.2. FGF-2: An Essential Regulator in Skeletal Tissue

FGF-2, also known as basic fibroblast growth factor, is a canonical FGF that belongs to the FGF-1 subfamily. It is encoded by *FGF-2*, on chromosome 4 [129], and is a wide-spectrum angiogenic, mitogenic and neurotrophic factor expressed by many types of cells in both adult and developmental stages [130]. It is a regulator of proliferation [131], migration [132], differentiation [122,129,133], cell survival [134], and stemness in human stem cells [135,136]. Apart from its key role in angiogenesis, FGF-2 is a major player in skeletal development, bone formation, and fracture repair [137,138,139]. During embryogenesis, it is a strong mesodermal inducer, and its receptors are strongly expressed in developing bones [140,141]. Throughout life, it is constantly expressed in osteoblasts and stored in the ECM [142]. Taken together, these properties make FGF-2 an attractive molecule for clinical and pharmaceutical applications in bone regeneration.

FGF-2 is highly expressed in bone tissues, with several isoforms due to alternative start codons for FGF-2 mRNA translation initiation [143]. The high molecular weight FGF-2 (HMW FGF-2) isoforms (24, 23, and 22 kD) are localized in the nucleus, whereas low molecular weight FGF-2 (LMW FGF-2) (18 kD) is cytoplasmic, and membrane associated. This differential intracellular trafficking also reflects a difference in function. Specifically, the LMW FGF-2 promotes osteoblast differentiation and mineralization via the activation of the Wnt pathway [144] and via synergistic actions with bone morphogenetic protein 2 (BMP-2). Indeed, endogenous FGF/FGFR signaling is a positive upstream regulator of the BMP-2 gene in calvarial osteoblasts [145]. In contrast, HMW FGF-2 acts in the nucleus as a transcriptional factor that upregulates the expression of genes associated with impaired mineralization, such as *SOST* and *FGF-23* [146].

Experimental in vitro evidence regarding FGFR-2 mutations associated with craniosynostosis syndromes highlights the major contribution of FGF-2 to bone cell fate. The genetic inactivation of *FGFR-2* causes reduced osteoblast proliferation and increased osteopenia, while *FGFR-1* has been associated with stage-dependent regulation of osteoblast proliferation and differentiation [147,148,149].

Exogenous FGF-2 rescued reduced bone nodule formation by upregulating the Wnt/β-catenin pathway in FGF-2-/- osteoblast cultures. [150,151]. Furthermore, when two FGFR-2 mutants from the Apert and Crouzon syndromes were expressed in immature osteoblasts, both inhibited osteoblastic differentiation but also increased apoptosis [152] downregulating many Wnt targets and inducing SRY-box 2 (SOX-2), a transcription factor that maintains the undifferentiated state of cells [153] (Figure 4). Furthermore, in a line of murine mesenchymal progenitor cells, induced overexpression of FGFR-2 led to increased cell proliferation and osteogenic differentiation through ERK1/2 by FGFR-2 signaling [154]. Inversely, primary calvarial osteoblasts from *Fgf*2-/- mice showed reduced BMP-2 induced periosteal bone formation [155]. In more mature cells, ERK1/2 activation by FGF-2 enhances acetylation and stabilization of RUNX-2, a key transcription factor involved in osteoblastogenesis and bone formation [114,156,157]. In addition, FGFR-2 signaling is crucial for the induction of apoptosis when osteoblasts are well differentiated [158,159], an important step for bone homeostasis. These results underline the dual temporal role of FGF-2 signaling in bone development, this protein promoting proliferation in immature osteoblasts and differentiation in mature ones. Furthermore, it should be underlined that since FGF-2 can bind equally to FGFR-1 and FGFR-2, differential activation of one of them may be responsible for signal transduction towards the proliferation of progenitors or towards osteogenic differentiation in post-proliferating cells [154].

The important dual role of FGF-2 in bone formation is saliently demonstrated by transgenic mouse models. Conditional deletion of *fgf-2* in mice yields a skeletal dwarfism phenotype and reduced bone formation [147]. Furthermore, *fgf-2* haploinsufficient mice are characterized by generalized osteopenia [151] and *fgf-2* knockout mice display greatly reduced trabecular plate-like structures and loss of connecting rods [160]. This reduced bone formation is due to defective osteoblast differentiation and alteration of progenitor cell lineage commitment, FGF-2 deficiency resulting in increased bone marrow adipogenesis and reduced osteogenesis [161]. Nonetheless, overexpression of *fgf-2* in mice also gives rise to skeletal defects, including a shortening and flattening of long bones, with a decrease in osteoblast differentiation, impaired bone formation, and moderate macrocephaly [162,163].

## 4. FGF-2 in Craniofacial Bone Tissue Engineering

In recent decades, tissue engineering has emerged as a new biomedical field with advanced approaches for tissue regeneration and healing [164]. It was initially defined as “the application of the principles of biology and engineering to the development of functional substitutes for damaged tissue” [165]. Bone tissue engineering needs to restore distinct functions: structural (e.g., bone, cartilage), barrier- and transport-related (e.g., blood vessels), and/or biochemical and secretory (e.g., hematopoiesis, calcium metabolism). 

Tissue engineering is based on the combination of three basic tools: cells, biomaterials, and suitable biochemical and physical factors, seeking to mimic the physical and functional properties of the natural tissue creating a tissue-like structure [166,167]. The last factor to consider is the presence of exogenous chemical and mechanical stimuli, such as soluble growth and differentiation factors (e.g., BMP, FGF-2, VEGF and TGF-β), and mechanical forces. These factors can be incorporated into a construct during scaffold fabrication itself or included in the culture medium to facilitate the survival, proliferation, and differentiation of the implanted cells and their integration into the host. This section focuses on FGF-2 as a good candidate for craniofacial bone tissue engineering by acting on both angiogenic and osteogenic processes. 

### 4.1. FGF-2: An Exogenous Factor In Vitro

Exogenous FGF-2 administration in vitro has multiple effects on bone cell fate. Firstly, it has been shown that FGF-2 maintains the osteoblast precursor proliferative state [142,168]. Its anabolic effect is evident through the stimulation of bone marrow stem cells, sustaining their osteogenic potential by maintaining the cells in an immature state with a fibroblast-like morphology expressing less alkaline phosphatase (ALP) [169]. DPSCs, stem cells from human exfoliated deciduous teeth (SHEDs) and stem cells from apical papilla (SCAPs) treated with FGF-2 express stemness-related markers including STRO-1 and CD-146 [170,171,172]. Recently, it has been reported that DPSCs/SHEDs displayed increased and prolonged proliferation upon FGF-2 treatment in vitro, together with delayed type 1 collagen expression [173]. On the other hand, human calvaria bone cells grown in mineralizing medium for several weeks with FGF-2 showed no stimulation of proliferation, suggesting that mature bone cells do not respond to FGFs’ mitogenic signal [174]. In fact, the decrease in intrinsic proliferation potential of human mesenchyme-derived progenitor cells with age was partially attributed to a reduction in FGF-2 expression, as observed in elderly humans [175]. In line with this, there is evidence that the efficacy of FGF-2 in inducing bone formation might be maximized if targeting younger cells, such as juvenile osteoblasts [176].

Concomitant with prolonged stemness, the abolition of mineralization by FGF-2 has been shown in various stem cell lines. Shimabukuro et al. [177] demonstrated that treatment of human DPSCs with FGF-2 increased their migration and proliferation ability but also impaired mineralization. On the contrary, if hDPSCs were only FGF-2 pretreated for a short period of time but left to differentiate under normal osteogenic conditions, ALP activity and nodule formation increased. Likewise, other studies confirm that pretreatment with FGF-2 during the proliferation phase leads to increased ALP activity, formation of mineralized nodules and expression of dentin sialoprotein and dentin matrix protein 1 (DMP-1) in hDPSCs [178]; dentin sialophosphoprotein (DSPP) and bone sialoprotein (BSP) expression in immature adult rat incisor dental pulp cells [179]; BSP, osteocalcin (OCN), osteopontin (OPN) and matrix Gla protein (MGP) in cementoblasts [180]; and hyaluronan in hDPSCs [181]. Nauman et al. [182] showed that in vitro administration of FGF-2 to rat osteoprogenitor cells accelerated the mineralization process through alkaline phosphatase and OCN expression and lowered the phosphate threshold needed for successful bone nodule formation. These observations suggest that FGF-2 enhances cell growth at early stages, a step that is crucial for accelerated differentiation at later time points. On the other hand, the spatiotemporal patterns of FGF signaling in vivo may differ from those found in culture conditions. Another possibility is that cell responses to FGF signaling in vivo are determined by, or are coordinated with, signaling from other cytokines in situ, such as BMPs and WNT [75,183].

### 4.2. FGF-2: An Exogenous Factor In Vivo

FGF-2 alone or in conjunction with other molecules and in combination with several types of scaffold has been assessed in various preclinical models for craniofacial bone regeneration. The rationale of FGF-2 administration is, first, its direct impact on bone cell proliferation and differentiation and, second, its pro-angiogenic action at the site of bone regeneration.

FGF-2 seems equally beneficial when tested in bone regeneration models of intramembranous or endochondral ossification. FGF-2 can be expressed by differentiating osteoblasts at sites of intramembranous ossification or by growth plate chondrocytes [128]. Its administration in vivo promotes regeneration of cranial [184,185], and periodontal [186,187,188] bone defects, as well as being associated with a shorter timeline of craniofacial bone repair [173]. Indeed, it has also been shown that FGF-2 was able to partially restore the lost cancellous bone mass in the ovariectomized rat [138]. The addition of FGF-2 significantly increased central defect bone filling in aged mice, leading to qualitatively superior bone formation. This suggests that FGF-2 is a good candidate for boosting bone regeneration in areas with impaired angiogenic potential or small numbers of native osteoprogenitor cells [189]. Similar benefits in angiogenesis and bone formation have been found in critical size defects in rat calvaria [190,191]. Furthermore, controlled delivery of FGF-2 in combination with a low dose of BMP-2 improved aged murine calvaria bone defect healing as compared to treatment with BMP-2 alone and the use of bone substitute impregnated with BMP-2 and FGF-2 promoted periodontal regeneration in non-human primates [189,192].

Vascularization plays a crucial role in bone tissue engineering seeking to replace large tissue losses due to trauma, surgery, or other clinical scenarios where spontaneous bone repair is not feasible. Indeed, using a radiotracer, Collignon et al. observed a correlation between early angiogenesis assessed by positron emission tomography and bone formation determined by micro-computed tomography within mouse calvarial bone critical size defects [193]. Recently, Novais et al. found that FGF-2 priming of DPSCs/SHEDs boosted intramembranous bone formation in critical size calvaria defects in immunodeficient mice [173]. Indeed, priming these cells with FGF-2 greatly enhanced stem cell early proliferation leading to increased bone regeneration concomitant with the expression of mineralization markers such as OPN, DMP1, or ALP.

These results suggest the importance of vascularization in bone regeneration. In fact, upon implantation in vivo, a major challenge is the maintenance of cell viability in the bone graft core, which critically depends on rapid invasion by host blood vessels. A functionally perfused vascular network will ensure oxygen and nutrient transport and waste removal. In this regard, endothelial cells play a key role in tissue regeneration and remodeling, since they can facilitate the recruitment of osteoprogenitors and immune cells, through the secretion of osteogenic factors, such as BMPs [194]. A recent study, with involvement of our research group, has shown implantation of a pre-vascularized scaffold network engineered in vitro to be a promising strategy for promoting blood supply deep into the graft, relying on inosculation with the host vasculature [195]. In particular, it has demonstrated the importance of grafting a mature microvascular network, displaying perivascular recruitment through the PDGF-BB pathway and basement membrane remodeling, taking advantage of the angiogenic properties of DPSCs/SHEDs and allowing self-assembly of endothelial cells into capillaries. Interestingly, we previously reported that the subcutaneous implantation of tissue-engineered constructs seeded with DPSCs/SHEDs primed with FGF-2, greatly enhanced vascularization within constructs thanks to the capacity of DPSCs/SHEDs to constitutively secrete hepatocyte growth factor (HGF) [170]. We have also demonstrated that FGF-2 is instrumental in promoting both VEGF and HGF secretion by DPSCs/SHEDs. Indeed, recruited stem cells participated in the deposition of vascular basement membrane and vessel maturation in athymic nude mice demonstrated the importance of in vitro production of mature microvasculature for improving cell-based therapies [195].

### 4.3. Human Clinical Applications of FGF-2 in the Craniofacial Area

The observations described above have been replicated in human clinical studies more recently.

Indeed, rhFGF-2 is already used in the clinical treatment of orofacial tissues. Kitamura et al. [196] performed a double-blind randomized controlled trial with 253 patients receiving rhFGF-2 0.2%, 0.3% or 0.4% or placebo during surgical management of periodontal intrabony defects. At 36 weeks, all FGF-2-treated groups demonstrated significantly higher radiographic bone fill than the placebo group, 0.3% being the best concentration. In addition, secondary analysis in a subgroup of patients showed very low levels of FGF-2 in serum and no adverse effects were reported. A randomized controlled trial of 30 patients showed an improvement in pocket depth reduction and more clinical attachment gain compared to control sites [197]. Application of FGF-2 for the treatment of intrabony defects has been studied in another randomized controlled trial, where various concentrations of FGF-2 were used mounted in β-tricalcium phosphate scaffolds. At 6 months, patients treated with 0.3% or 0.4% rhFGF-2 showed 71% success for the combined outcome of attachment gain of 1.5 mm and bone fill of 2.5 mm compared to 45% success in the 0.1% FGF-2 and control groups [198]. A meta-analysis of studies using recombinant human FGF-2 for the treatment of deep intrabony periodontal defects demonstrated a clinical benefit of FGF-2 in terms of bone fill [199]. A more recent meta-analysis of six randomized controlled trials shows that administration of 0.4% rhFGF-2 yielded 22% higher bone fill of periodontal defects than control treatment, though this result was not statistically significant. It also indicated that the impact of the treatment was dose dependent, with higher FGF-2 concentrations producing better bone regeneration outcomes [200]. To date, however, there is still no consensus on the optimal dose or delivery scaffolding method for the use of FGF-2 in the field of bone regeneration.

### 4.4. FGF-2 as an Exogenous Factor: A Synthesis of Current Knowledge

It is evident from animal models and human clinical application studies that exogenous administration of FGF-2 is a promising method for accelerating craniofacial bone regeneration. Given the spatiotemporal effect of FGF-2 on mineralization, a key challenge is to determine the optimal application strategy. Parameters such as dose, length of exposure, administration mode, and type of scaffolding, as well as the origin and differentiation state of the stem cells employed for craniofacial bone tissue engineering appear to be crucial. These parameters are discussed in the following sections and summarized in Table 1.

#### 4.4.1. Dosage (In Vitro/In Vivo)

In vitro, FGF action is complex, and the biological effect of FGF-2 may depend on the dose, length and mode of exposure. Mouse bone chip outgrowth cells that were primed with FGF-2 (0, 0.0016, 0.016 or 0.16 ng/mL) demonstrated dose-dependent expression of mesenchymal markers, suggesting dose-dependent anabolic action of FGF-2 on proliferation [201]. Human mesenchyme-derived progenitor cells from cancellous bone were harvested from young and old patients and cultured under various FGF-2 concentrations (0.0016, 0.016, 0.16 and 1.6 ng/mL) for 4, 24, 28 and 72 h. Responsiveness regarding proliferation was dose and age dependent, with the proliferation rate diminishing with age [202]. Despite the inhibitory effect on differentiation and mineralization, the addition of FGF-2 to culture medium maintains stemness in SHEDs and embryonic stem cells [203], and it has been shown to be necessary to maintain the cells in a pluripotent state [204]. Indeed, continuous treatment of cultured osteoprogenitors with 10 ng/mL of rhFGF-2 over 2 days significantly reduced expression of alkaline phosphatase, osteopontin and collagen I expression, though RUNX-2 mRNA levels were not altered, indicating that cells treated with FGF-2 retain their osteogenic commitment [145]. Threshold doses vary between studies (e.g., Varkey et al. demonstrated that concentrations higher than 2 ng/mL inhibit proliferation and differentiation of bone marrow mesenchymal cells) [205].

In vivo, several studies have been conducted to assess the performance of FGF-2 in various administration modes in vivo with varying dosages depending on the animal model and bone defect configuration. A single local injection of FGF-2 directly at the site of interest has shown to increase periosteal bone formation in murine calvaria [206] and facilitate the healing of bone fracture and segmental bone defect in rats [137], rabbits [207,208], dogs [209], and non-human primates [210,211]. Kamo et al. compared the effect of a single local injection with that of cyclical injections of FGF-2 on a cancellous bone defect in the femoral condyle of rabbits. Only the high-dose single injection (1.2 μg/μL), and not the low-dose single injection (0.4 μg/μL) or cyclical injections (0.4 μg/μL, 3 times), significantly increased cancellous bone volume as measured by bone histomorphometry [212]. These results indicate that the effect of local injections of FGF-2 on cancellous bone regeneration is greater at the very early stage of bone healing. Thus, the positive effect on proliferation is useful, but only if it is temporary and regulated. A similar conclusion can be drawn from the results of Novais et al. with short FGF-2 priming of SHEDs/DPSCs before their implantation in the calvaria defect [173].

On the other hand, some animal studies have shown the efficacy of FGF-2 with a prolonged administration mode. The anabolic effect of FGF-2 was observed with daily systemic administration for 2 weeks in a rat bone defect model [213,214,215]. Lane et al. found that FGF-2 treatment for 14 days (1 mg/kg/rat) was associated with the development of new trabecular elements, but with the withdrawal of FGF-2 injections, the new trabeculae were rapidly lost thorough accelerated resorption [216]. Furthermore, it has been reported that continuous local infusion of bFGF using an osmotic pump is able to shorten the consolidation phase of limb lengthening in rabbits [217].

#### 4.4.2. Scaffolding and Stabilization by Administration Mode

The dose is not the only parameter to consider. The administration mode influences the stability of FGF-2. Indeed, the inherent instability of this protein in aqueous solutions necessitates its delivery via biomimetic scaffolds that can stabilize and maximize its biological activity for a defined period of time [218]. FGF-2 exhibits a short half-life of 12 h in vivo due to degradation by proteolytic enzymes but also because of its instability as soon as it is thawed [219,220]. A crucial factor for the sustained release of FGF-2 is the resorption rate of the scaffold in which the growth factor is impregnated. A well-documented method is the use of gelatin hydrogels for the fabrication of FGF-2-loaded scaffolds for tissue regeneration applications since they can mimic the manner in which FGF-2 is stored in the ECM. Indeed, when FGF-2 in solution was directly injected ectopically in mice, the vascularization process remained unchanged, whereas the incorporation of FGF-2 into a gelatin hydrogel greatly enhanced neovascularization. Moreover, hydrogels with less water content (77.5% vs. 95.9%) were more efficacious in sustaining FGF-2 release due to their slower resorption rate [221]. Biodegradable gelatin hydrogel incorporating rhFGF-2 has been developed successfully in Japan and shown to restore bone [219,222,223].

FGF-2-mediated tissue regeneration has also been tested with chitosan/collagen scaffolding. FGF-2 controlled release by a chitosan/fucoidan complex hydrogel is presumed to immobilize it, prolong its biological half-life time and protect it from inactivation by heat or proteolysis. After injection of this hydrogel into an ectopic murine model, significant neovascularization was observed, attributed to both slow diffusion of FGF-2 and controlled biodegradation of the hydrogel [183]. Similar findings were reported with the use of heparin/protamine water-insoluble microparticles for FGF-2 delivery [224]. Various other heparin-mimicking molecules have been tested for FGF-2 administration, such as heparin mimetic peptide nanofibers [225], sulfated peptides [226], sulfonated dextrans [227] and polysulfonated polymers [228,229]. Combining poly (lactic-co-glycolic acid) (PLGA) and poly (vinyl alcohol) (PVA) to produce FGF-2-loaded microspheres has also shown to sustain the release and stability of FGF-2 for up to 4 days in a culture of human embryonic stem cells. Furthermore, FGF-2-loaded polycaprolactone (PCL) microspheres have been reported to enhance angiogenesis in vivo [230], and the use of microspheres constructed from combined alginate/collagen hydrogels yields a scaffold that provides controlled release of FGF-2 and enhances angiogenesis [231].

Radomsky et al. showed that a single local injection of FGF-2 in a hyaluronan gel, an ECM component, significantly promoted fracture healing of the fibulae in baboons, as evidenced by increased callus formation and mechanical strength [211]. Tabata et al. reported that FGF-2 incorporated into gelatin hydrogel induced bone formation at the site of a skull defect in non-human primates [184]. More recently, Murahashi et al. [232] have developed multi-layered FGF-2-loaded poly L-lactic acid nanosheets. Their subcutaneous application allowed the sustained release of loaded rhFGF-2 for about 2 weeks and enhanced bone regeneration upon implantation at critical size femoral murine defects. Controlling rhFGF-2 stability and delivery will make it possible to adapt the dose and length of exposure necessary to the bone defect to be repaired.

#### 4.4.3. BMP-2: An Interesting Cytokine for Combined Treatments

FGF action is complex, and the biological effect of FGF-2 may depend on its interaction with other cytokines. The combination of FGF-2 and BMP-2 has already been tested at various concentrations and kinetics actions. A recent in vitro study suggests that sheep BMSCs supplemented with 20 ng/mL of FGF-2 and 100 ng/mL of BMP-2 may be a feasible cellular therapy for bone regeneration [233]. In vivo, the association of 10 μg of FGF-2 and 10 μg of BMP-2 in transfected human MSCs yielded a significantly increased bone regeneration of critical size calvarial defects of nude mice [234]. Indeed, implanted calcium phosphate ceramic tubes loaded with rat marrow MSCs preconditioned with both FGF-2 and BMP-2 yielded better bone formation than FGF-2 or BMP-2 treatment alone [235]. In addition, a novel biomimetic coating scaffold (calcium phosphate/polyelectrolyte multilayer (bCaP-PEM)) capable of sequential delivery indicated that FGF-2 delivery followed by BMP-2 increased bone regeneration in adult mouse calvarial bone defects more than delivery of BMP2 alone [201]. The combined action of FGF-2 and BMP-2 is also dose dependent. Notably, scaffolds loaded with 2 μg of BMP-2 on collagen disks enhanced osteoinduction when FGF-2 was in the range of 16–400 ng but inhibited with 10 or 50 μg of FGF-2 [236]. Similarly, using implants in rats, Takita et al. [237] found that 100 ng of FGF-2 was able to enhance BMP-2 (0.8 μg), inducing ectopic bone formation, whereas a higher dose of FGF-2 (10 μg) exerted an inhibitory effect. Higher doses may keep cells in an undifferentiated state for a longer time, this negatively impacting mineralization [173]. A possible mode of action is that high doses of FGF-2 do not increase the expression of BMP receptors, whereas low doses of FGF-2 strengthen bone formation via BMP-2 signaling as shown by increased Smad 1 expression, a major downstream effector in the BMP signaling pathway [238].

## 5. Discussion, Conclusions and Perspectives

Bone is a constantly evolving mineralized and vascularized connective tissue that can be repaired by simple immobilization in the case of non-displaced fractures. During major trauma, infection or cancer; however, bone loses its capacity to self-repair.

Tissue engineering is considered a therapy of the future, making it possible to clinically overcome the many limitations of current autologous graft therapies (notably, the limited quantity of tissue available and risk associated with several surgical sites on a single patient). The combination of biomaterials, cells and molecules of interest may allow great advances at the clinical level by stimulating the integration of grafts through vascularization and mineralization. The close relationship between blood vessels and bone cells has been demonstrated in studies on skeletal malformation, such as craniofacial dysmorphology.

FGF-2, by virtue of its proliferative, pro-angiogenic and pro-osteogenic properties, is one of the molecules studied in this line of research. This growth factor is a ubiquitous molecule present from the embryonic stage and throughout life and is involved in the formation of the ECM. It plays an important role in the homeostasis, repair and metabolism of bone tissue by regulating the proliferation and differentiation of osteoblasts, accelerating the healing of fractures and the repair of skeletal defects.

The effects of FGF-2 differ depending on the type and stage of cell differentiation. Some results appear to be contradictory but can be explained by differences in dose, mode of administration and lengths of exposure, as well as by the models used in in vitro or in vivo studies, each of which has certain biases.

It appears that high dose and/or long-term treatment inhibit bone regeneration. This could be explained by a positive effect on proliferation, which remains essential at the beginning, but must be temporary, and therefore regulated. Thus, low dose and/or short-term treatment may provide the best conditions for bone regeneration. There is a need to explore further the positive or negative effects on bone regeneration of other parameters, such as the type of culture medium used and any supplements added to boost the osteogenic effect (ascorbic acid, dexamethasone, β-GP, etc.). The way in which FGF-2 is delivered should also be considered, since it may affect bone repair, potentially leading to unwanted side effects. Moreover, the delivery mode seems to have a non-negligible impact on the stability of this cytokine, and therefore, must be carefully planned and tested before clinical use. Identifying the ideal dose and how to deliver it over a given time within a specific repair time frame remain the key challenges in this type of tissue-engineered therapy.

## Figures and Tables

**Figure 1 cells-10-00932-f001:**
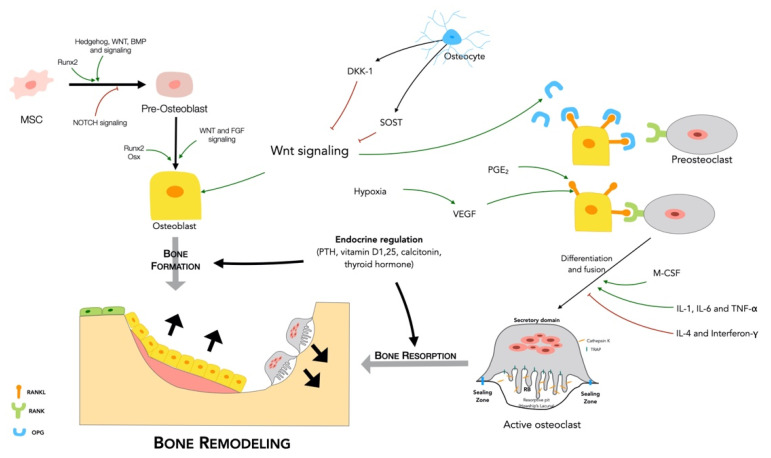
Bone remodeling regulation can be paracrine or endocrine. Several factors participate in paracrine regulation including cytokines (IL-1, IL-6, TNF-alpha, IL-4 and interferon-gamma), PGE2, VEGF, and hypoxia, as well as bone cells. There are three main cell types involved: osteoblasts, osteocytes and osteoclasts. Osteoblasts, which differentiate from mesenchymal progenitors thanks to certain proteins (Runx2, Osx and Wnt) and FGF signaling pathways, are responsible for bone formation. They can also become osteocytes able to regulate osteoblastogenesis through production of inhibitors (DKK-1 and SOST), that inhibit Wnt signaling. Lastly, osteoclasts, involved in bone resorption are activated through RANK-RANKL-OPG signaling pathway cross-talk. Whenever there is a need for bone resorption, osteoblasts and osteocytes express RANKL on their surface, and this then binds to RANK in osteoclast precursors, activating their differentiation. OPG is then secreted to stop bone resorption binding to RANKL blocking the possibility of RANK-RANKL binding and preventing bone resorption. Once activated, mature osteoclasts bind to the bone matrix, becoming polarized. Their cytoskeleton organizes into actin rings forming the sealing zone, which provides an isolated acidic microenvironment, to dissolve minerals and digest the selected bone matrix thanks to the ruffle border (RB). After resorption, the osteoclasts endocytose the degraded collagen fragments, and the calcium and phosphate released are then transported through the cell and liberated at the functional secretory domain before being released into the bloodstream. Bone formation and resorption are also influenced by endocrine regulation. Various factors may be involved, for example, PTH, 1,25(OH) Vitamin D, calcitonin and thyroid hormone.

**Figure 2 cells-10-00932-f002:**
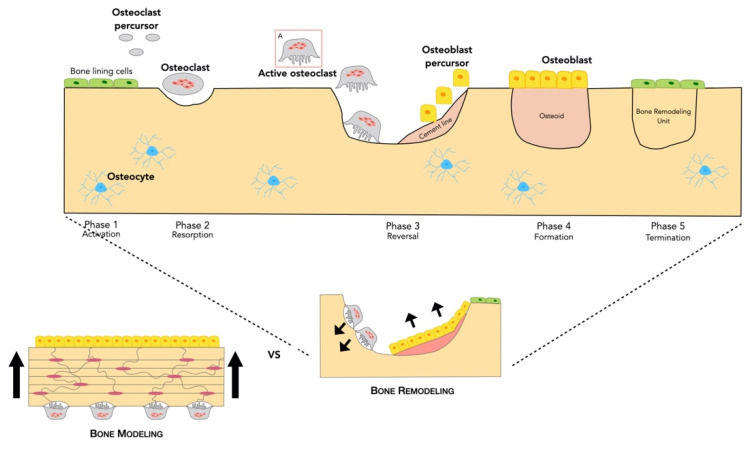
Bone metabolism: Modeling vs. Remodeling—While bone modeling implies a change in bone shape or size since resorption and formation occur independently at distinct sites: osteoblasts secrete osteoid matrix in the opposite site where osteoclasts resorb bone. Bone remodeling involves the resorption and formation of bone, one after the other, at the same site to replace old and/or damaged bone by newly formed bone. An initiating remodeling signal, such as hormonal or mechanical signal, is detected by the bone, inducing the release of paracrine factors that lead to retraction of the bone lining cells which exposes the bone surface, allowing recruitment of osteoclast precursors from the capillaries directly into the basic multicellular unit. MSC-F and RANKL, secreted by osteocytes, induce recruitment of precursor cells of hematopoietic lineage, initiating their differentiation to multinucleated osteoclasts. The differentiated attached osteoclasts rearrange their cytoskeleton to adhere to the bone surface, decreasing the pH to as low as 4.5, this dissolving the bone mineral. Once resorption is finished, the osteoclasts go through apoptosis. After resorption, mononuclear cells are recruited to remove collagen fragments from the surface, and then new osteoblasts begin collagen deposition, forming what is known as osteoid matrix, until the cavities are filled. Osteoblasts produce new bone, and some of them become buried within the newly formed bone matrix turning into osteocytes with their extensive canalicular network connecting them to the bone surface lining cells, osteoblasts and other osteocytes. The osteoid mineralizes, and the bone enters into a quiescent phase.

**Figure 3 cells-10-00932-f003:**
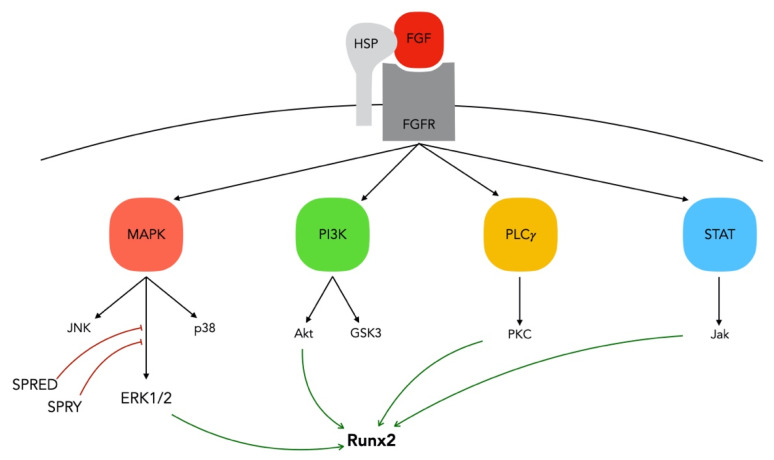
FGF/FGFR signaling—FGFs can bind to FGFRs with the help of heparan sulfate, a co-factor, and thereby induce their biological effects through activation of four major signaling pathways: RAS-MAPK-ERK1/2, PI3K-AKT-GSK3, PLCγ-PKC, and STAT-Jak.

**Figure 4 cells-10-00932-f004:**
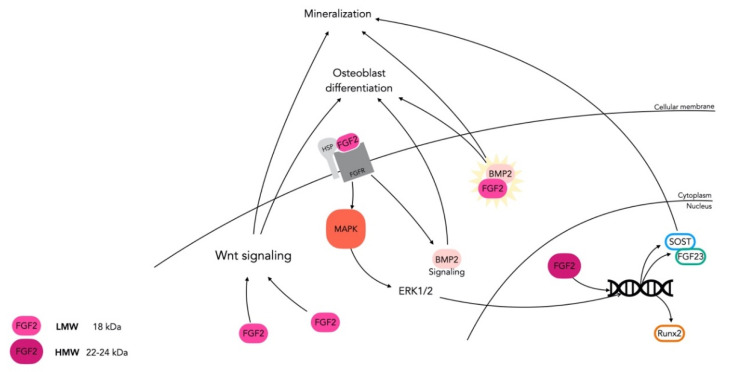
FGF/FGFR signaling in bone—FGF-2 is highly expressed in bone tissues. There is a high molecular weight (HMW) form, located in the nucleus, that acts as a transcriptional factor, and upregulates the expression of *SOST* and *FGF-23* responsible for inducing mineralization. The low molecular weight (LMW) form is cytoplasmic or membrane associated. The latter can promote osteoblast differentiation and mineralization through the Wnt pathway, BMP-2 signaling, or synergistic action with BMP-2. By activation of FGFR signaling, LMW FGF-2 also activates MAPK-ERK/2, which acts as a transcriptional factor that upregulates mineralization genes such as *RUNX2*.

**Table 1 cells-10-00932-t001:** Overview of FGF-2 used in craniofacial studies arranged by research type: dose, timing, administration mode and scaffolding.

Study	Application	Dosage and Timing	Administration and/or Scaffolding	Results
in vitro studies				
Gromolak et al. 2020	Ovine bone marrow MSCs	FGF-2 (20 ng/mL) alone or in combination with BMP-2 (100 ng/mL)	In culture medium	Osteogenic differentiation induced by BMP-2 is amplified with FGF-2 supplementation. FGF-2 alone boosted proliferation of smaller cells, but without osteoblast-like structures in culture and decreased expression of osteogenic genes.
Li et al. 2014	Murine calvarial osteoblasts	FGF-2 (1, 10, 20, 60 ng/mL)	In culture medium	Doses ≤10 ng/mL yielded higher cell proliferation Doses >10 ng/mL decreased proliferation Increased mineralization at all doses
Sukarawan et al. 2014	Stem cells from human exfoliated deciduous teeth (SHEDs)	FGF-2 (10 ng/mL)	In culture medium	FGF-2 maintains cell stemness
Ou et al. 2010	Murine calvarial and femur osteoprogenitor cells Human cancellous bone osteoprogenitor cells from young and old patients	rhFGF-2 (0.0016, 0.016, 0.16, or 1.6 ng/mL) 4, 24, 48, and 72 h	In culture medium	Accelerated proliferation at all doses FGF-2 induced proliferation diminished with age
Varkey et al. 2006	Rat bone marrow cells	FGF-2 (2, 10, 50 ng/mL) and BMP-2 (50, 150, 500 ng/mL) over 3 weeks	In culture medium	Accelerated mineralization at 10 ng/mL but reduced at 50 ng/mL of FGF-2 Synergistic role of FGF-2 and BMP-2 in old rat cells
Animal models				
Novais et al. 2019	Critical calvarial bone defects in nude mice	FGF (10 ng/mL) over 72 h	SHEDs in dense collagen matrices in osteogenic culture medium	Enhanced bone formation in calvarial critical size defect
Wang et al. 2019	Mandibular defects in non-human primates	FGF-2 (0.25 μg/μL)	Calcium phosphate cement for BMP-2 carrier PGA gel for FGF-2 carrier	Promotion of periodontal regeneration
Anzai et al. 2016	2-wall periodontal defects in Beagle dogs	FGF-2 (3 mg/mL) vs. vehicle	Cellulose solution Direct injection in the defect	FGF-2 promoted regeneration in alveolar bone, cementum and periodontal ligament.
Charles et al. 2015	Calvarial bone defects in old mice	FGF-2 (5 ng) and BMP-2 (2 μg)	Collagen hydroxyapatite discs	Enhanced bone filling in the central bone defect area when BMP-2 was supplemented with FGF-2
Akita et al. 2004	Calvarial defects in nude mice	FGF-2 (2.5 ng/mL) and BMP-2 (50 ng/mL) vs. FGF-2 alone, BMP-2 alone or vehicle	Transfected human MSCs in gelatin sponge carrier	Combination of FGF-2 and BMP-2 showed the most advanced bone formation within the defects
Clinical studies				
Cochran et al. 2016	Patients with periodontal intrabony defects	rhFGF-2 (0.1%, 0.3%, 0.4%) or no application	β-TCP	Increased clinical attachment gain and bone fill at concentrations of 0.4% and 0.3%
De Santana et al. 2015	Patients with periodontal intrabony defects	rhFGF-2 (4 mg/mL) vs. no application	Sodium hyaluronate gel	Enhanced clinical parameters of wound healing compared to negative control
Kitamura et al. 2001	Patients with periodontal intrabony defects	rhFGF-2 (0.2%, 0.3%, 0.4%) or placebo	3% hydroxypropyl cellulose gel	At 36 weeks, all defects showed bone fill except placebo 0.3% dose had the best radiographic outcomes

## Data Availability

Not applicable.

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
