# Peer review of "The Potential of FGF-2 in Craniofacial Bone Tissue Engineering: A Review"

_cells, 2021, doi:10.3390/cells10040932_

Round 1

Reviewer 1 Report

Thank you for the opportunity to review this manuscript.

The authors present a review article on FGF-2 in bone tissue engineering. Large bone defects are a unsolved problem in all areas of musculoskeletal surgery. The authors address an important topic in this field. The article is divided in two parts: first the description of bone physiology and second the characteristics of FGF-2 for bone regeneration.

The article is clear and well-structured. The length is appropriate.

Author Response

We thank the reviewer for the time spent in revising our manuscript and for his/her positive comments on our work and hope he/she enjoyed reading it.

Reviewer 2 Report

The article looks in to basic understanding of bon turnover and especially the influence of FGF2 in bone remodeling. Especially the part on FGF 2 and its therapeutic possibilities are interesting.

Major concerns:

The manuscript should be extensively shortened. Most of the section on bone physiology is interesting but very well-known and could therefore be shortened

The description of the bone cells could be more stringent and shortened. Many of the descriptions of metabolites such as RANK etc are better and in more detail described in the very nice section “Regulation of bone remodeling”, therefore it is not necessary to describe this amongst the description of bone cells.

Also in other parts of the manuscript some information is repeated several times. Look for repetitive sequences in the text to shorten the manuscript

Minor concerns:

In Fig 2 the different cell types should be mentioned in the figure, osteocytes are not mentioned- Also the bone modeling should be better described

1,25(OH) Vitamin D is the correct description of the active vitamin D.

RUNX-2 should be written in full length the first time mentioned (Runt-related transcription factor 2)

Author Response

We thank the reviewer for the time spent in revising our manuscript and for the valuable comments provided, which greatly contributed to improve the manuscript.

Considering his/her major comments, we agree with the reviewer on the fact that part A summarizes already well-known data, so we made many changes, especially in the Bone Cells part by shortening this section while retaining the essentials for resituating the action of FGF-2 in the bone physiology. We hope that this reviewer will be satisfied with our major changes.

Concerning the minor revisions, we changed the size and format of all the cells in Figure 2, allowing a better view of the osteocyte at the bottom left of the figure and expanded the legend of Figure 2 to include elements of the main text that we have removed to make it shorter (line 140-144). We also thank reviewer 2 for correcting us on the writing of Vitamin D (1,25(OH) Vitamin D) and have corrected the manuscript accordingly (line 103 and 200), and the full length of RUNX-2 is now located (line 70).

We hope that this reviewer will be satisfied with our clarifications and changes highlighted in yellow in our revised manuscript.

Reviewer 3 Report

The submitted review entitled “The potential of FGF-2 in bone tissue engineering: a review” is divided in 3 parts where the authors first describe aspects of bone physiology, the role of fibroblast growth factor-2 (FGF-2) in bone homeostasis and finally discuss its use in bone tissue engineering (BTE).

The review is well structured and written. Bellow you can find minor comments and suggestions.

- The authors provide a good panel of bone structures and mechanisms of bone formation and remodeling in the physiological context. Since the focus of the review is to bring up the potential use of FGF in BTE treatments, it would be interesting to introduce the functions and regulation of FGF in an injured bone environment. Would bone formation and remodeling signaling occur in healthy bone and during healing of normal and large bone fractures in the same way?

- In section C2, the use of FGF-2 to treat large bone defects is reported. Much of the review presents in vivo studies in cranial bones. The results of the FGF-2 treatment are similar in studies performed in long bones?  Since mechanism of healing might be slightly different, there would be differences in the influence of FGF in intramembranous and endochondral ossification? Some sentences could be added in this context.

-Table 1 provides a summary of studies using FGF-2, in which the most recent studies are from 2019 and specifically in the section for in vitro studies, only 4 studies are listed and the most recent are from 2014. No studies have investigated the influence of FGF-2 in vitro more recently or there were some exclusion/inclusion criteria to select studies for this review?

Minor comment:

In supplementary file, the table S1 is cut at the end.

Author Response

We thank the reviewer for the time spent in revising our manuscript and for the valuable comments provided, which greatly contributed to improve the manuscript.

Considering the interest of the FGF-2 pathway for injured bone repair, we thank the reviewer for this relevant suggestion. Although we have included some representative studies of osteoporotic and fracture models, the pathological bone is an extremely broad subject. Here, we mainly focused this review on craniofacial and oral bone defects, which undergoes mostly intramembraneous ossification. This point is now better explained in the legend of Appendix Table 1 (lines 739-743). To avoid misunderstandings, we added the term “craniofacial” in the headings and in the manuscript (lines 2, 23, 51, 412, 427, 469, 478, 521, 548, 552, 563, 746) and removed all the data in Table 1 not involved in Craniofacial area but also in Part C2 and C3 accordingly. We just kept the general osteoporotic study [138] to highlight the craniofacial study [189]  we found in aged murine calvaria bone defect, which could explain the interest of FGF-2 by its ubiquitous action (line 486).

We thank reviewer 2 for his comment regarding Table 1. We agree that the in vitro studies are not the most recent in the field of FGF-2 for bone tissue engineering. Indeed, we do not use a systematic research on multiple databases, we selected only the most representative studies we found with explicit details on FGF-2 dosage concentrations or different time points. However, we updated our research and found a relevant more recent one, published in December 2020 [233]. We also realized that we forgot two in vivo studies, and added them in Table 1 [187, 234]. Indeed, we decided yet to delete the word “recent” from the table’s legend to remain rigorous in our terminology. We also rearrange and complete the BMP-2 section in Part C3 to further clarify the various points found and specify our statements (lines 650-672).

Regarding the minor comment, we have added a missing bar marking the end of the Table S1.

We hope that this reviewer will be satisfied with our clarifications and changes highlighted in yellow in our revised manuscript.

Reviewer 4 Report

This manuscript by Novais et al. entitled “The potential of FGF-2 in bone tissue engineering: a review” reviews the literature and highlights the role of bFGF in bone pathophysiology as well as its applications in bone regenerative bioengineering.

This is a comprehensive and well written review describing in detail the roles of bFGF in skeletal tissue. The authors are explicit with regards to the parts selected to be included in the manuscript (bone physiology, FGF in bone and tissue engineering) and included the essential references covering these fields. It is very interesting that they included both animal and human studies for the use of bFGF in the treatment of bone disorders. The illustrations are well structured, and tables are useful for the reader.

Some minor comments:

Line 72, which are channels. Delete or rephrase.

Lines 72-74, please rephrase.

Line 90, please rephrase. The authors probably mean ossification or mineralisation.

Lines 145-146. Please rephrase. Bone remodeling is continuous throughout life course. It is correctly written in the next sections.

Line 167. This is called chondrocyte-to-osteoblast trans-differentiation.

Author Response

We thank the reviewer for the time spent in revising our manuscript and for the valuable comments provided, which contributed to improve the manuscript.

Concerning the minor comments, we agreed with all of them and deleted “which are channel” line 72 and rephrase lines 145/146, now lines 123/124: “Bone modelling occurs primarily during growth and development in childhood, although it can also appear after the skeleton has matured”.  

One of the reviewer asked us to shorten Part A, so we had to delete some of the lines (lines 72-74) affected by some of this reviewer comments and move line 167 to the legend of the Appendix Table S1 (lines 742-746): “There are two different processes of bone modelling: endochondral ossification, involving cartilage as an intermediate stage, and intramembranous ossification, the chondrocyte-to-osteoblast trans-differentiation, which involves direct differentiation of MSCs into osteoblasts. The former is typical for long bones, while the latter is the main mechanism for the development of flat bones, such as those of the craniofacial skeleton”.

We also rephrased (line 90, now 83) as requested by “Other types of cells, such as chondrocytes are found within the bone. These are derived from pluripotent MSCs, and secrete type II collagen, participating in the endochondral ossification process”.

We hope that this reviewer will be satisfied with our clarifications and changes highlighted in yellow in our revised manuscript.

Round 2

Reviewer 2 Report

Non